# An Improved Real-Time Contrasts Control Chart Using Novelty Detection and Variable Importance

**Kwang-Su Shin, In-seok Lee and Jun-Geol Baek \***

School of Industrial Management Engineering, Korea University, Seoul 02841, Korea;
shinkscon@korea.ac.kr (K.-S.S.); dark7710@korea.ac.kr (I.-s.L.)
**\*** Correspondence: jungeol@korea.ac.kr; Tel.: +82-2-3290-3396

**Abstract:** Fault detection and isolation are important tasks in statistical process control. A real-time contrasts (RTC) control chart converts the statistical process-monitoring problem to the real-time classification problem, thus outperforming traditional monitoring techniques. An RTC assigns a class to reference data and the other class to a window of real-time contrasts. However, RTC control charts often fail to detect abnormal states when both normal and abnormal data exist together in the window. To enable more rapid detection of an improved RTC control chart, this paper proposes a multivariate process monitoring system with an improved RTC control chart. Although previous RTC control charts proposed by other studies outperform the original RTC chart, it is still difficult to detect an abnormal state when normal and abnormal data exist together. To overcome this problem, this paper proposes an RTC control chart using novelty detection and variable importance with random forests. Novelty detection and variable importance were used so that fault can be detected when the control limit could not be exceeded despite the abnormal state. The proposed method extracts representative data in the sliding window and adds the extracted data to the window to quickly detect the abnormal state. Experiments demonstrate the proposed method to outperform the original RTC chart.

**Keywords:** real-time contrasts (RTC); control chart; novelty detection; variable importance; fault detection; multivariate exponentially weighted moving average (MEWMA)

---

## 1. Introduction

Recently, developments in manufacturing have led to an increase in the amount of multivariate data collected from the manufacturing processes. Because of the importance of multivariate data, multivariate statistical process control (MSPC) is nowadays essential to enable the simultaneous monitoring of multivariate variables [1,2]. The multivariate exponentially weighted moving average (MEWMA) control chart [3], multivariate cumulative sum (MCUSUM) control chart [4], and $T^2$ control chart [5] are examples of representative multivariate process controls. Typical multivariate process control charts assume a normal distribution of data, estimate the parameters in the normal state, and use them as a statistic for control charts. However, in actual manufacturing processes, hardly any data satisfy the normal distribution. If the normal distribution is assumed, the performance is degraded. To overcome these problems, several approaches have been proposed to classify the state of multivariate process control charts as normal or abnormal by using machine-learning techniques. Although the performance is better than that of a conventional multivariate process control chart, once the model learned in phase I is generated, the decision boundary does not change, and the performance may deteriorate. To improve this problem, there have been some studies to generate artificial contrasts [6–8] and a method called real-time contrasts (RTC) has been proposed [9]. RTC learns a new classifier as the sliding window progresses. This method calculates the statistics based on sequential classifications. Because it learns a new classifier in real time, it offers the advantage

of better reflecting the characteristics of the observations in Phase II in the control chart. RTC uses random forests as the classifier to calculate the statistics. This classifier offers the advantage of being able to check variables that are considered the cause of abnormal observations in real time. However, a problem associated with RTC is that the monitoring statistics of the original RTC have discrete value because these monitoring statistics are computed based on the binary classification probabilities of several decision trees [10].

To overcome these limitations, distance-based RTC has been proposed. This method uses a support vector machine (SVM) and kernel linear discriminant analysis (KLDA) as a classifier to calculate the statistics [11]. Attempts have been made to improve the performance of the classifier by applying weighted voting to random forests to improve the performance of RTC while using random forests to maintain the advantage of being able to identify anomalous cause variables in real time [10]. Most previous attempts to improve RTC performance have used other classifiers instead of random forests. However, these studies could not identify the abnormal cause variable in real time. Some studies have attempted to use random forests to improve the detection ability of the control chart by improving the performance of the classifier. RTC creates contrasts using a sliding window. However, when both normal and abnormal data exist together in the sliding window, the sensitivity to detect abnormalities is decreased. In addition, in an abnormal state, when some normal observations are input to the sliding window, they can act as noise and degrade the RTC performance. Also, although the process is out of control, when normal and abnormal data are mixed and normal data is intermittently observed in the sliding window, the out-of-control state cannot be detected quickly. Therefore, in this study, we suggest a means of improving the performance of the RTC control chart. We improved the classification performance of the RTC control chart by enhancing it with a variable importance chart using random forests, novelty detection using SVM, and MEWMA.

The proposed method consists of two phases. In Phase I, novelty detection is performed using the reference data. Through this novelty detection, it is possible to determine the extent of the real-time contrasts from the reference data. We also check the variable importance of the reference. We set the maximum value of the variable importance of the reference data as the threshold of the contrasts. Second, in Phase II, if the contrast variable importance value is higher than the threshold of variable importance, it can be assumed to be in an abnormal state even though the monitoring statistics do not exceed the threshold of the monitoring statistics. In this case, to increase the statistics of the RTC control chart in the abnormal state, we use the novelty detection to determine the location of the data in the contrasts that vary from the normal state. Using novelty detection, we align the data in the contrasts near the boundary with the initial direction of the contrasts. Conversely, we align the data far from the boundary with the recent direction of the contrasts. Then, we use MEWMA to obtain data that is representative of the sliding window. The existing MEWMA method assigns a greater weight to recent data. However, in this paper, it was used to assign a greater weight to suspected abnormal data. If the extracted data are attached to the contrasts, an abnormality can be detected more quickly than with the conventional method. This paper consists of the following sections. Section 2 introduces the original RTC control chart and then Section 3 describes the newly proposed method. In Section 4, we compare the performance of the original RTC control chart with that of the newly proposed method. Section 5 concludes this paper and discusses further studies to be undertaken in the future.

## 2. Real-Time Contrast Control Chart

In this section, we describe the original RTC control chart [9] that uses random forests as a classifier.

### 2.1. Real-Time Contrast (RTC)

The characteristics of RTC control chart differ from those of a traditional control chart, specifically, in the application of the sequential hypothesis test. An RTC control chart is a type of sequential classification. This method defines reference data and real-time contrasts to classify each other in real time. The RTC control chart generates monitoring statistics by comparing reference data $S_0$ and

real-time contrasts $S_w$. $S_0$ consists of the normal state of Phase I. The size of $S_0$ is $N_0$, while $S_w(t)$ is the contrasts containing the most recently observed data. $S_w(t)$ is constructed by applying the concept of a moving window. This involves setting the window size $N_w$ and constructing a contrast by moving the window to include the most recent observations. In the RTC control chart, $S_0$ is set to class 0 while $S_w(t)$ is set to class 1 to learn the classifier. The classifier learns whenever the new $S_w(t)$ is created. If $S_w(t)$ consists of the normal state observations, it is difficult to classify each class. However, when $S_w(t)$ is composed of abnormal state observations, it utilizes the feature whereby the classification of decision boundary is clear and easy to classify. The probability that an arbitrary observation $x_i$ until time $t$ is classified as class $k$ ($k$ = 0, 1) is expressed as $\hat{p}_k(x_i)$.

$$p(S_0, t) = \frac{\sum_{x_i \in S_0} \hat{p}_0(x_i|t)}{N_0}, \text{ for } x_i \in S_0 \tag{1}$$

$$p(S_w, t) = \frac{\sum_{x_i \in S_w(t)} \hat{p}_1(x_i|t)}{N_w}, \text{ for } x_i \in S_w(t) \tag{2}$$

It can be seen that Equations (1) and (2) are affected by the size of the reference data and the size of the sliding moving window, respectively. Furthermore, $N_0$ should be set such that it is much larger than $N_w$. Consequently, $p(S_0, t)$ is sensitive and its stable detection performance is better than $p(S_w, t)$. For this reason, in this study, we used only $p(S_0, t)$. Figure 1 illustrates the process of the RTC control chart.

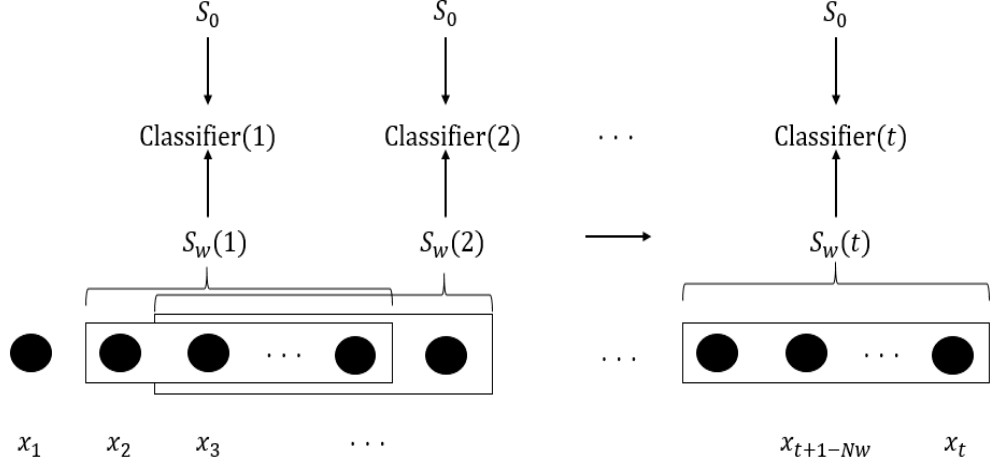

**Figure 1.** Framework for real-time contrast (RTC) control chart.

## 2.2. Control Limit

In a MSPC chart, the control limit is used to distinguish between normal and abnormal states. First, the control limit is set through statistics in the normal state after determining an acceptable type-I error ($\alpha$). Second, the type-II error ($\beta$) in the abnormal state and the abnormal detection ability are determined by the control limit. Therefore, to minimize the type-I and type-II errors, it is important to set a highly efficient control limit to ensure abnormality detection. In the process control chart, $RL_k$ (run length) is the amount of data observed prior to the control limit first being exceeded. The average run length (*ARL*) is expressed by the following equation.

$$ARL = \frac{1}{R} \sum_{k=1}^{R} RL_k \tag{3}$$

$$ARL_0 = \frac{1}{\alpha} \tag{4}$$

$$ARL_1 = \frac{1}{1 - \beta} \tag{5}$$

$R$ in Equation (3) is the number of repetitions. In this study, a performance evaluation was carried out by fixing $ARL_0$ and the control limit was set to $ARL_0 \cong 200$.

## 2.3. Random Forests

The random forests algorithm is very effective and useful for both prediction and classification problems. It was originally developed by Breiman [12–14]. A decision tree-based random forests is a robust model for outliers and is able to handle various data types, interaction between variables, and nonlinearity. It is also used in RTC control charts because of its ability to effectively obtain a probability estimation. Random forests should set two parameters as a classifier that constructs an ensemble using the bagging (bootstrap aggregation) of a large number of decision trees. First, the number of decision trees constituting an ensemble should be determined. Second, the number of randomly chosen variables should be determined. In the case of decision trees, the overfitting problem does not occur even if a sufficiently large number of decision trees is set. For this reason, in this study, the number of decision trees was set to 500. Also, the number of randomly selected variables was set to the square root of the number of variables [9–11]. Observations that are not used for bagging in the learning process of an individual decision tree are called out-of-bagging (OOB) observations. These are used to estimate the probability of anomalies and enable the fault identification of anomalies. The probability of the OOB observations for each observation $x_i$ belonging to class $k$ being correctly predicted by the original class is defined as

$$\hat{p}_k(x_i) = \frac{\sum_{j \in OOB_i} I[\hat{y}(x_i, \ t_j) = k]}{|OOB_i|}, k \ = \ 0, \ 1 \tag{6}$$

In addition, the predicted class is 0 when $\hat{p}_0(x_i)$ less than control limit and 1 otherwise. An indicator function $I(\cdot)$ returns 1 for true or 0 for false for the argument and $\hat{y}(x_i, t_j)$ represents the predicted class. Here, $t_j$ is the $j^{th}$ decision tree among the decision trees in the random forests. $OOB_i$ is the set of decision trees that do not use observation $x_i$ in the learning process using bagging. An additional consideration is the class-imbalance problem. Because $N_0$ is set such that it is very large compared to $N_w$ when the classifier learns, there is the problem of predicting the class of all the observations to the class of $N_0$. To overcome this problem, the downsampling of $N_0$ can be applied. If downsampling is performed in the process of learning an individual decision tree, it is possible to improve the classification speed of random forests by improving the computation speed and decreasing the correlation between decision trees [12–14]. In this study, the size of the downsampling is the same as the sliding window size.

## 2.4. Fault Isolation Using Variable Importance

In the original RTC control chart, the cause of a fault can be analyzed by evaluating the importance of variables in the classifier when a fault detection alarm occurs [9]. In this section, we introduce a method for determining the importance of variables proposed by Breiman [12–14]. The Random forests algorithm is used to measure the significance contained in the decision tree. A thorough search of every node of every tree in the random forests scored $m$ selected variables. As a result, we can implicitly consider the importance of each variable to the model with impurity reduction as a measure of relative importance. Variable importance is the degree of impurity reduction. Breiman [12–14] proposed the application of variable importance by using the Gini index of impurity. Equation (7) is described in terms of the Gini index in a random forest.

$$Gini(v) = \sum_{i=1}^{c} r_i(1 - r_i), \tag{7}$$

where $c$ is the number of classes, $v$ is a variable as a node, $r_i$ is the ratio of the total data to class $i$ and $s_v$ means split in node $v$. The impurity reduction $\Delta Gini(s_v, v)$ at each node $v$ reflects the ratios $w_L$ and $w_R$ in that they are proportional to the number of the data for each child node from the parent node $v$.

$$\Delta Gini(s_v, v) = Gini(v) - (w_L Gini(v_L) + w_R Gini(v_R)) \tag{8}$$

The impurity reduction $\Delta Gini(s_v, v)$ is the difference between the impurity score of node $v$ and the weighted average of impurity scores of children nodes of node $v$. Then, in variable $X_k$, the variable importance $V(X_k)$ is the average of the impurity reduction $\Delta Gini(s_v, v)$ for all trees.

$$V(X_k) = \frac{1}{ntree} \sum_{t=1}^{ntree} \sum_{v \in D_t} \Delta Gini(X_k, v) \tag{9}$$

## 3. Proposed Method

We propose an improved RTC control chart with novelty detection and variable importance. In the case of the original RTC control chart, the reference data defined as normal state was used only as a comparison group for the contrast data. However, in this study, we not only used the reference data as a comparison group, but also took the reference data's maximum value of the variable importance as the control limit of the real-time variable importance. The proposed method consists of Phases I and II, as shown in Figures 2 and 3.

In Phase I, there are three steps: deciding the variable importance threshold of the real-time contrasts, deciding the decision boundary using novelty detection, and deciding the RTC control limit. We use the reference data to determine the novelty detection's decision boundary, extract the variable importance of the real-time contrast's threshold, and set the control limit. In this paper, SVM is used to perform novelty detection. Especially, the radial basis function (RBF) kernel is known to perform well in novelty detection, so this study used the RBF kernel [15].

As shown in Figure 4, we have confirmed that the distribution of the contrasts differs from the reference data. Figure 4 shows when an abnormal state is caused by $X_2$. As shown in Figure 4, some data are separated from the normal state distribution. However, it can be seen that there are data that are very close to the distribution of the normal state even though it is in an abnormal state. For this reason, when an abnormal state occurs in the RTC control chart, abnormal and normal data may be mixed in the contrasts. In such a case, the performance of $ARL_1$ may deteriorate.

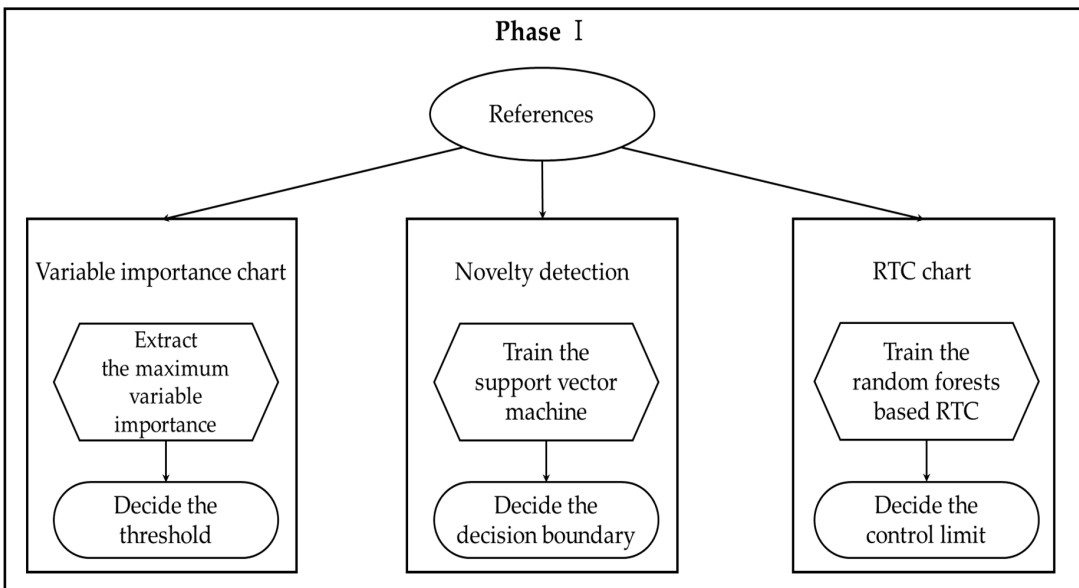

**Figure 2.** Phase I of the proposed method.

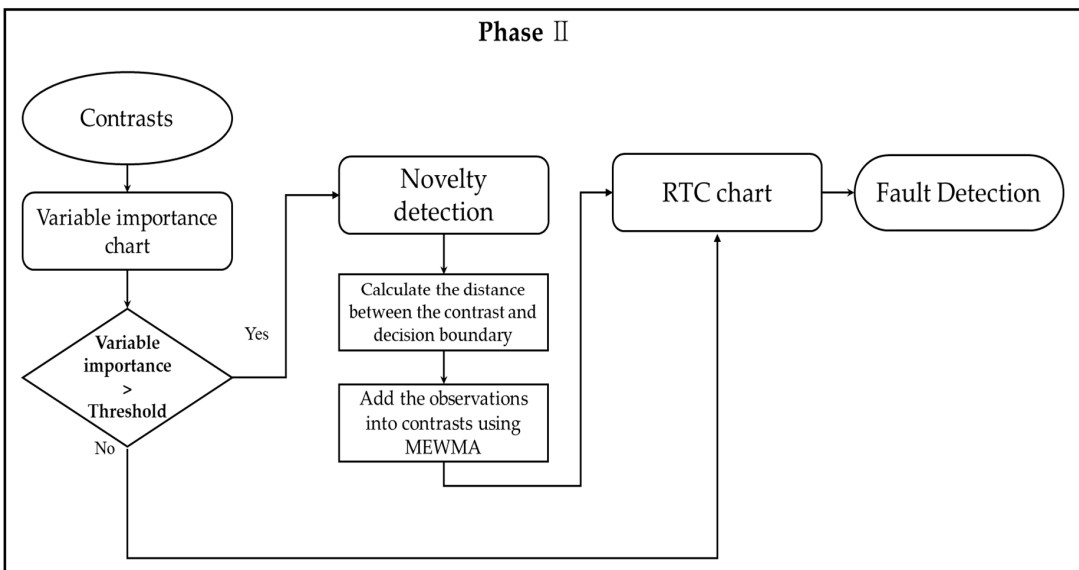

**Figure 3.** Phase II of the proposed method. MEWMA: multivariate exponentially weighted moving average.

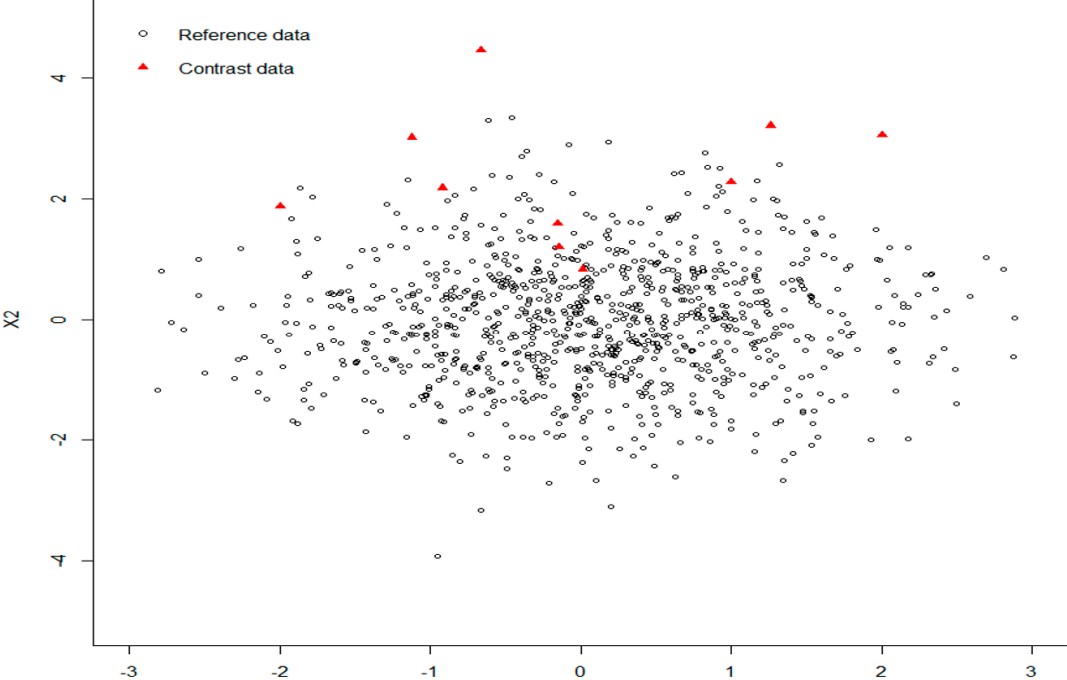

**Figure 4.** Abnormal state data in sliding window.

Figure 5 shows how the data is shifted to the variable $X_2$, which differs from the other variables. Unlike in Figure 4, Figure 5 shows that the cause of the abnormality can be clearly detected. For this reason, we set the threshold for the variable importance of the reference data as a criterion for determining whether the proposed algorithm works. However, monitoring statistics will often be falsely replaced with variable importance charts. Therefore, we try to set the threshold and improve the performance by using the variable importance chart of normal data. In Phase II, the information obtained from Phase I is utilized as soon as the variable importance of the contrast exceeds the threshold. We introduce an improved real-time contrast using novelty detection and variable importance and describe the proposed method in Sections 3.1 and 3.2 in more detail.

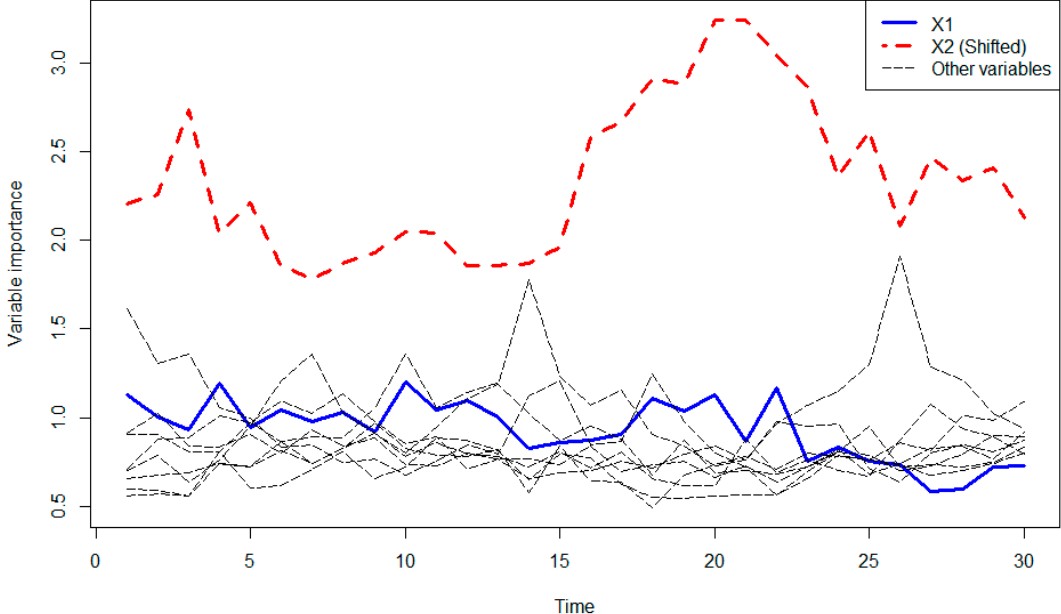

**Figure 5.** Variable importance; number of trees = 500, window size = 10.

### 3.1. Phase I

In this paper, the proposed method consists of two phases. In Phase I, we performed novelty detection, checking the variable importance using the reference data, and deciding the RTC control limit. Novelty detection involves modeling the normal state, thus enabling the detection of any divergence from normality. In other words, if any data conflicts with the decision boundary based on the normal data, we calculate the distance whether it is inside or outside the boundary. This distance information will be used in Phase II. In this study, the algorithm to be used for novelty detection is SVM. SVM can be used as a nonlinear decision boundary for classification. To perform nonlinear classification, it is necessary to map the given data to the high-dimensional feature space. With the development of the multivariate control chart, several studies used the kernel distance to reflect the high dimension [16–19]. Previous studies have used a kernel distance to create a multivariate control chart [18]. There have been attempts to apply not only high-dimensional data using kernel distances but also control charts. The RBF kernel exhibits good performance in one-class classification [18]. In this paper, SVM was used because of its good ability to handle nonlinear data as well as high-dimensional data using the RBF kernel. As shown in Figure 5, we can see that we can quickly identify the anomalies with the variable importance chart. For this reason, we set the maximum value of the variable importance of reference data to the threshold of the variable importance of real-time contrasts, thus triggering the proposed method. Finally, we set $ARL_0 \cong 200$ to fix the RTC control limit as described in Section 2.2. In Phase II, the proposed method is implemented when the variable importance in the sliding window exceeds the maximum value obtained in Phase I.

### 3.2. Phase II

In Phase II, the proposed method is implemented when the variable importance exceeds the threshold value. However, if it does not exceed the threshold, it will perform the same real-time monitoring as the original RTC control chart. First, if the variable importance is exceeded, we check the distance between the contrasts and the boundary obtained through the novelty detection. Then, we align the data in the contrasts. Finally, we use MEWMA to create data that represent the contrasts. MEWMA is a logical extension of the univariate EWMA and is defined as follows:

$$Z_i = \theta x_i + (1 - \theta)Z_{i-1}, \ i = 0, \ 1, \ 2, \ \dots \tag{10}$$

$Z_0$ is applied as the target or starting value. Hunter [20] proposed a guideline value for $\theta$ (where $\theta$ is the weight given to the most recent observation $x_i$) of 0.2 to 0.3 for data processing. MEWMA was designed to be capable of weighting the data by time [21–23]. However, we converted MEWMA to weighted data far from boundary of novelty detection. In this paper, we gave more weight to data that is close to the abnormal state as a result of sorting data in the novelty detection by using MEWMA. Then, we attached the extracted data to real-time contrasts. We attempted to solve the problem of the performance of $ARL_1$ being deteriorated by the slowing of the fault detection despite the occurrence of the abnormal state by adding the extracted data to the contrasts.

## 4. Experiments

In this section, we compared the performance of the proposed method with the existing method. To verify the performance, we changed the experimental conditions such as the dimensions and shift sizes. We generated normal and abnormal observations to compare the detection performance of the original RTC control chart with the $ARL_1$ for a given $ARL_0 \cong 200$. The results of the previous study show that the best performance is obtained when the window size = 10 [8]. Therefore, we did not compare the performance based on the window size.

### 4.1. Data Description and Experimental Design

The experiment was performed by generating the data for a normal distribution with mean of 0 for which the covariance matrix is the identity matrix. The size of the comparative group data was set to 2000 ($N_0$ = 2000), consisting of normal observations, and $N_w$. is the size of the sliding window. The data size of the comparative group was set to 2000 and consisted of normal observations. In the case of abnormal state data, data with a mean shift of the normal data was generated. The data used to evaluate the experimental results consisted of 20 normal observations and 20 abnormal observations. The performance index of the multivariate process control chart, *ARL*, was used as the evaluation index. We used $ARL_1$ as the performance measure where $ARL_0$ is 200, with the normal data being repeated 1000 times. Abnormal observations were set as mean changes, and experiments were conducted for one variable (Case 1) and three variables (Case 2) as the causative variables. The shift size was computed by the non-centrality parameter $\lambda$, defined as follows.

$$\lambda = \sqrt{\delta^T \Sigma_X^{-1} \delta} \tag{11}$$

where $\delta$ is the magnitude vector of the mean change. $\Sigma_X^{-1}$ is a diagonal matrix where the diagonal elements corresponding to the shifted variable are 1 and the remaining elements are 0. Table 1 summarizes the conditions used in the experiment.

**Table 1.** Experiment condition.

| Description | Parameter | Value |
| --- | --- | --- |
| Size of reference data | $N_0$ | 2000 |
| Window size | $N_w$ | 10 |
| Non-centrality parameter | $\lambda$ | 1.0, 2.0, 2.5, 3.0 |
| Weighting constant of MEWMA | $\theta$ | 0.3 |
| Dimension of observation vectors | $p$ | 10, 50, 100 |

### 4.2. Experimental Result

The experimental results are shown in Tables 2–4. Table 2 represents the result obtained when the size of dimension was fixed to 10, and we compared the performance of the proposed method according to the shift size. Table 2 lists the values of $ARL_1$ and the standard deviation. Tables 2–4 indicate that, as the amount of the shift decreases with the proposed method, the level of the performance increases.

In addition, in both cases, we can see that an improvement in the performance is attained with the proposed method.

In Tables 2–4, we fixed the window size and compared the performance by different shift size and dimensions. We found that the performance improvement increases with the dimensions.

**Table 2.** Experimental results, dimension size = 10, window size = 10.

| Shift Size | $ARL_1$ (Std. Error) | | | |
| :---: | :---: | :---: | :---: | :---: |
| | Case 1 | | Case 2 | |
| | Original RTC | Proposed RTC | Original RTC | Proposed RTC |
| 1.0 | 9.03 | **8.11** | 9.94 | **8.81** |
| | (0.27) | (0.18) | (0.1) | (0.13) |
| 2.0 | 8.64 | **7.90** | 8.79 | **8.07** |
| | (0.36) | (0.41) | (0.42) | (0.33) |
| 2.5 | 7.92 | **7.29** | 7.99 | **7.58** |
| | (0.29) | (0.33) | (0.24) | (0.27) |
| 3.0 | 7.12 | **6.86** | 7.37 | **6.92** |
| | (0.3) | (0.33) | (0.21) | (0.28) |

**Table 3.** Experimental results, dimension size = 50, window size = 10.

| Shift Size | $ARL_1$ (Std. Error) | | | |
| :---: | :---: | :---: | :---: | :---: |
| | Case 1 | | Case 2 | |
| | Original RTC | Proposed RTC | Original RTC | Proposed RTC |
| 1.0 | 10.88 | **9.72** | 11.22 | **10.17** |
| | (0.33) | (0.19) | (0.24) | (0.17) |
| 2.0 | 9.95 | **8.81** | 10.57 | **8.49** |
| | (0.47) | (0.41) | (0.42) | (0.33) |
| 2.5 | 9.20 | **8.55** | 9.18 | **8.71** |
| | (0.18) | (0.18) | (0.19) | (0.11) |
| 3.0 | 8.54 | **8.04** | 8.39 | **8.25** |
| | (0.38) | (0.11) | (0.47) | (0.27) |

**Table 4.** Experimental results, dimension size = 100, window size = 10.

| Shift Size | $ARL_1$ (Std. Error) | | | |
| :---: | :---: | :---: | :---: | :---: |
| | Case 1 | | Case 2 | |
| | Original RTC | Proposed RTC | Original RTC | Proposed RTC |
| 1.0 | 12.70 | **10.31** | 12.66 | **11.98** |
| | (0.09) | (0.16) | (0.37) | (0.12) |
| 2.0 | 12.09 | **10.01** | 12.27 | **10.05** |
| | (0.36) | (0.41) | (0.42) | (0.33) |
| 2.5 | 11.78 | **10.12** | 11.52 | **9.97** |
| | (0.56) | (0.26) | (0.49) | (0.18) |
| 3.0 | 11.25 | **9.90** | 11.30 | **9.02** |
| | (0.13) | (0.14) | (0.15) | (0.41) |

Figure 6 shows the monitoring statistics for the original and proposed RTC control charts. We set the normal and the abnormal states for 20 times. This is an excerpt from one of the experiments (dimension size = 100, shift size = 2, window size = 10) in Case 1. From the instant that the abnormal state was established, the monitoring statistics were found to increase, and it was confirmed that the proposed method can detect the fault more quickly than the existing method.

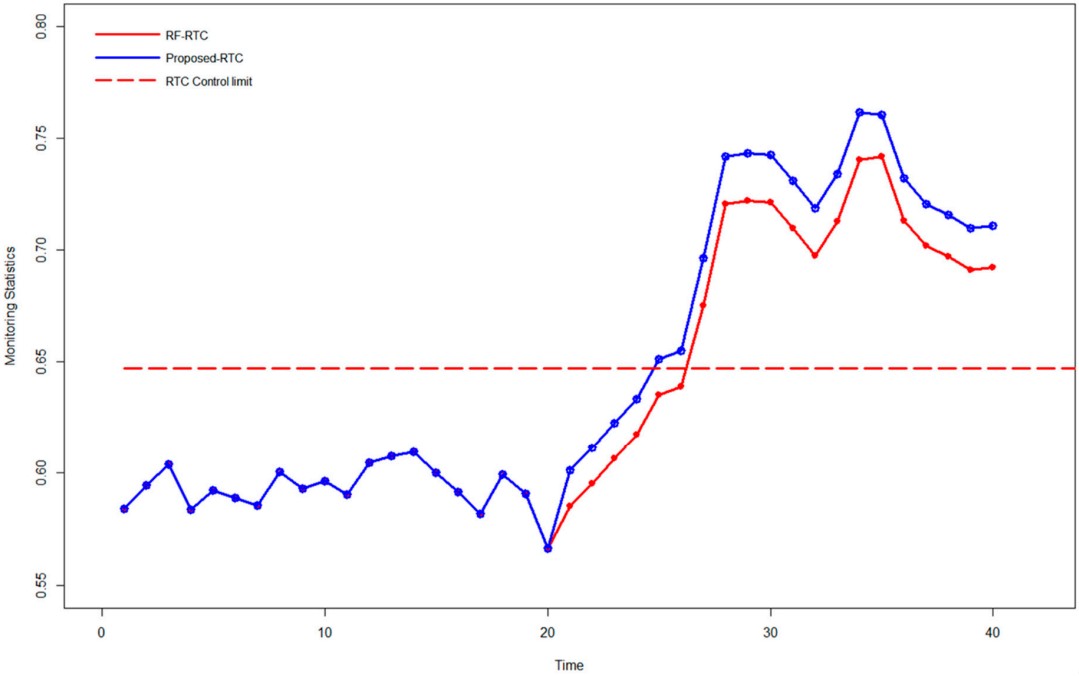

**Figure 6.** Comparison of monitoring statistics.

## 5. Conclusions

In this study, we proposed an improved real-time contrasts method using novelty detection and variable importance. The proposed method used the variable importance chart and novelty detection. In the variable importance chart, we used the maximum value of the variable importance in the reference data as a threshold. When the threshold of the variable importance in the real-time contrasts is exceeded, we use novelty detection to sort the data in the contrasts. In this sorting of the contrasts, weighted data is extracted through MEWMA. The extracted data is added to the contrasts and then the RTC control chart is activated. We proposed a method that improves the performance of $ARL_1$ when both normal and abnormal data exist together in the contrasts. In other words, when the process is in an abnormal state, the performance degradation problem, which had an inflow of data similar to normal state data in the contrasts, was resolved through the proposed method. Experimental results show that the performance of the proposed method is better than that of the original RTC control chart. In the future, we will examine how to make the proposed method operate flexibly, depending on the dimensions of data, the degree of data shift, and the window size.

**Author Contributions:** K.-S.S. proposed the idea and carried out the experiments. I.-s.L. assisted with the numerical modeling and analysis. J.-G.B. validated the proposed method and guided the research. All the authors have read and approved the final manuscript.

**Funding:** This work was supported by the National Research Foundation of Korea (NRF) grant funded by the Korean government (MSIT) (NRF-2016R1A2B4013678). This work was also supported by the BK21 Plus program (Big Data in Manufacturing and Logistics Systems, Korea University) and by Samsung Electronics Co., Ltd.

**Conflicts of Interest:** The authors declare no conflict of interest.

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
