# Peer review of "An Improved Real-Time Contrasts Control Chart Using Novelty Detection and Variable Importance"

_applsci, doi:10.3390/app9010173_

Reviewer 1 Report

Please update your references and use more recent papers.

Please update your conclusion as I found minor grammatical errors.

Author Response

We would like to thank you for your thoughtful review. 

Please see the attached PDF file for descriptions of how we addressed your comments.

Thank you again for your helpful comments.

Reviewer 2 Report

This paper presented an approach of RTC control chart using novelty detection and variable importance with random forests. 

Although the authors used experimental results, they don't refer their repository or their kind.

I would also expect to see more experimental evaluation.

Some of their references in the text needs correction (e.g. lines 160 &165).

Editing of English language and style required (e.g. lines 157-158 &160).

Author Response

(The authors gave the same response as above.)
